# Enhanced Antioxidant Capacity of Puffed Turmeric (*Curcuma longa* L.) by High Hydrostatic Pressure Extraction (HHPE) of Bioactive Compounds

**DOI:** 10.3390/foods9111690

**Published:** 2020-11-18

**Authors:** Yohan Choi, Wooki Kim, Joo-Sung Lee, So Jung Youn, Hyungjae Lee, Moo-Yeol Baik

**Affiliations:** 1Department of Food Science and Biotechnology, Graduate School of Biotechnology, Institute of Life Science and Resources, Kyung Hee University, Yongin 17104, Korea; yo3247@naver.com (Y.C.); kimw@khu.ac.kr (W.K.); daishi8882@naver.com (J.-S.L.); 2Department of Food Engineering, Dankook University, Cheonan 31116, Korea; govl1990@naver.com

**Keywords:** turmeric, curcumin, puffing, high hydrostatic pressure, antioxidant

## Abstract

Turmeric (*Curcuma longa* L.) is known for its health benefits. Several previous studies revealed that curcumin, the main active compound in turmeric, has antioxidant capacity. It has been previously demonstrated that puffing, the physical processing using high heat and pressure, of turmeric increases the antioxidant and anti-inflammatory activities by increasing phenolic compounds in the extract. The current study sought to determine if high hydrostatic pressure extraction (HHPE), a non-thermal extraction at over 100 MPa, aids in the chemical changes and antioxidant functioning of turmeric. 2,2-diphenyl-1-picrylhydrazyl (DPPH), 2,2′-azino-bis (3-ethylbenzothiazoline-6-sulphonic acid) (ABTS), and ferric reducing antioxidant power (FRAP) analyses were conducted and assessed the content of total phenol compounds in the extract. The chemical changes of curcuminoids were also determined by high performance liquid chromatography (HPLC). Among the three variables of ethanol concentration, pressure level, and treatment time, ethanol concentration was the most influential factor for the HHPE of turmeric. HHPE at 400 MPa for 20 min with 70% EtOH was the optimal extraction condition for the highest antioxidant activity. Compositional analysis revealed that 2-methoxy-4-vinylphenol was produced by puffing. Vanillic acid and ferulic acid content increased with increasing HHPE time. Synergistic effect was not observed on antioxidant activity when the turmeric was sequentially processed using puffing and HHPE.

## 1. Introduction

Turmeric (*Curcuma longa* L.) has been used widely as a spice, but its applications are limited due to its pungent flavor [1]. Many traditional medicines have utilized turmeric as a beneficial ingredient, and recent studies identified the major functional components, which are curcuminoids, essential oils, fixed oils, and various volatile oils including turmerone, alantone, and zingiberone [2]. Among these bioactive components, curcuminoids and their derivatives have been thoroughly investigated and reported to have anticancer, antibacterial, antioxidant, and anti-inflammatory properties [3]. Curcumin (1,7-bis-(4-hydroxy-3-methoxyphenyl-1,6-hepatadiene-3,5-dione), the major curcuminoid component, is a yellowish polyphenolic compound and has been used in many drugs and foods [4]. Curcuminoid degradation products, i.e., ferulic acid, vanillic acid, and vanillin, are also present in turmeric and have demonstrated antioxidant activity [5,6].

Various extraction methods, such as high hydrostatic pressure extraction (HHPE), microwave extraction (ME), supercritical fluid extraction (SFE), ultrasound, pulsed electric field (PEF), moderate electric field (MEF), etc., have been developed to enhance the yield or efficacy of bioactive components [7]. The application of supercritical carbon dioxide extraction on turmeric was optimized and turmerone was determined to be the major component [8]. In the case of supercritical carbon dioxide extraction of turmeric, high pressure (20–40 MPa) and low temperature (313–333 K) were the best extraction conditions for turmerone. Dandekar and Gaikar also reported a novel hydrotropy-based extraction method for selective extraction of curcuminoids from turmeric [9]. It was demonstrated that application of pressurized liquids at 10 MPa is an economic method for augmented extraction of curcuminoids [10]. Puffing, a simple processing method, was recently applied to turmeric resulting in increased antioxidant capacity in vitro [3]. Puffing uses relatively high heat and pressure and reportedly allows for increased antioxidant activity of natural resources, such as doraji, cacao beans, coffee beans, turmeric, and ginseng [3,11,12,13,14,15,16,17,18]. In addition, the Maillard reaction during the puffing process imparts a unique flavor by producing volatile substances, such as formic acid, acetaldehyde, formaldehyde, and glyoxal acid [19].

Among recent advances in food processing methods, high hydrostatic pressure (HHP) treatment is a non-thermal pasteurization method that uses pressure over 100 MPa [20,21,22,23]. HHPE is an effective method for extraction of heat sensitive materials. HHPE increases the yields and mass transfer rates of herbal products by cell wall breakdown as compared to the conventional extraction methods such as soxhlet, heat reflux, ultrasonic, microwave, and supercritical CO_2_ extractions [24,25,26,27,28].

While a significant number of papers has indicated the beneficial roles of turmeric in health as well as its active components of curcuminoids, its applications in processed foods are limited due to its insolubility in water resulting in low bioavailability and bioaccessibility [1,2,3,4,14]. In this regard, many approaches, including nano-particlization, encapsulation, emulsion formation, and bioconversion, to enhance the biofunctions of turmeric have been reported [2,11,12,13]. Since there is no report on the HHPE of turmeric, this study investigated the best HHPE conditions for extraction of turmeric to maximize the antioxidant activity. In addition, the combinational effect of puffing and HHPE on antioxidant activity and bioactive compounds of turmeric was also investigated.

## 2. Materials and Methods

### 2.1. Materials and Chemicals

Dried turmeric (moisture content 15.7 ± 0.1%, crude fat 6.4 ± 0.1, crude protein 7.7 ± 0.1, ash 6.5 ± 0.4) was kindly donated from Bibong Herb (Yangju-si, Korea) and kept in a deep freezer before use. Fermented ethanol, an extraction solvent, was provided by ethanol Supplies World Co. (Jeonju-si, Korea). Methyl alcohol, sodium carbonate, sodium hydroxide, and hydrochloric acid were purchased from Daejung Chemicals & Metals Co. (Siheung-si, Korea). Folin-Ciocalteu’s phenol reagent, 2,2′-azino-bis (3-ethylbenzthiazoline-6-sulfonic acid) diammonium salt (ABTS), 2,2′-azobis (2-amidino-propane) dihydrochloride (AAPH), 1,1-diphenyl-2-picrylhydrazyl (DPPH), gallic acid, catechin, ascorbic acid, HPLC analytical standard curcumin, demethoxycurcumin, bis-demethoxycurcumin, ferulic acid, and 2-Methoxy-4-vinylphenol were purchased from Sigma-Aldrich Co. (St. Louis, MO, USA). Aluminum chloride and sodium nitrite were purchased from Junsei Chemical Co., LTD (Tokyo, Japan). HPLC grade water, acetonitrile, and methanol were purchased from Honeywell Burdick & Jackson Inc. (Charlotte, NC, USA).

### 2.2. HHPE

HHPE was carried out using a high hydrostatic pressure unit (Suflux, Ilshin Autoclave Co., Daejeon, Korea). Extraction pressure, treatment time, and ethanol concentration were considered as the variable factors in this experiment according to Lee et al. [29]. Ground turmeric (5 g) and solvent (100 mL) were placed into a plastic pouch and hermetically heat-sealed. Following an application of high hydrostatic pressure at a specific condition, the solutions were vacuum filtered by using Whatman no. 2 filter paper (Cytiva, Marlborough, MA, USA) on a funnel and Kimble-filtering flask. The filtrate was subsequently stored at −20 °C for further experiments. It has been reported that pressure level and treatment time did not have a big impact on the HHPE of puffed ginseng [30]. Therefore, we wanted to confirm whether pressure level and treatment time had an impact on the HHPE of turmeric. In order to determine the best HHPE conditions for the highest antioxidant activity of turmeric, pressure, treatment time, and ethanol concentration were varied. First, pressure was varied (0.1 (atmospheric), 100, 250, 400, 550 MPa) with fixed treatment time (15 min) and ethanol concentration (70% ethanol) [26,29]. Second, treatment time was varied (5, 10, 15, 20, 30 min) with the best fixed pressure as previously determined (400 MPa) and ethanol concentration (70% ethanol). Last, ethanol concentration was varied (0, 20, 40, 70, 95% *v/v*) with the previously determined best pressure (400 MPa) and treatment time (20 min). Consequently, the best HHPE conditions were applied for the combinational effect of puffing and HHPE. 

### 2.3. Combination of HHPE and Puffing

Puffing of turmeric was carried out at 980 kPa as previously reported [3]. Briefly, 150 g of sliced and dried turmeric was mixed with 600 g of rice (1:4 *w/w*), which served as an excessive carbonization preventer at a high temperature and a heat-transfer medium [3,15,28]. The mixture was placed in a rotary gun puffing machine chamber and subjected to an incremental increase of internal pressure to 980 kPa by heating with a gas burner. Subsequently, puffing of the mixture was induced with a sudden pressure release by opening of the chamber door. The puffed turmeric was ground and HHPE was applied at 400 MPa for 20 min using a 70% ethanol concentration. The non-puffed with no HHPE sample was considered as the control sample. For extraction of non-puffed (control) and puffed turmeric, ground samples (5 g) in a 70% ethanol solvent (200 mL) were agitated by a magnet stirrer for 30 min at room temperature.

### 2.4. Extraction Yield

The extract was dried at 105 °C using a drying oven (HB-502M, HanBeak Scientific Co., Bucheon, Korea), and the extraction yield was determined by the following Equation (1) [3]:(1)$$\mathrm{Extraction}\;\mathrm{yield}\;\left(\%\right)=\frac{\left(w_{2}-w_{1}\right)}{A}\times \frac{E}{E^{\prime }}\times 100$$
where,
*A* = Weight of used sample (g)*E* = Total volume of extract (mL)*E*’ = Used volume of extract to be dried (mL)*w*_1_ = Initial weight of aluminum dish (g)*w*_2_ = Weight of aluminum dish and dried sample (g)

### 2.5. Radical-Scavenging and Ferric-Reducing Activities

Antioxidant activity of all turmeric extracts (ground turmeric, HHPE turmeric, puffed turmeric, puffed and HHPE turmeric) was determined using three different methods. In order to assess the radical scavenging activity, DPPH and ABTS radical scavenging activity were carried out. A FRAP assay was performed to determine the ferric-reducing activity. Ascorbic acid served as a standard, and the antioxidant activity of the extracts were determined in the unit of mg vitamin C equivalent (VCE)/g dried turmeric [31]. The radical scavenging and ferric-reducing activities of each extract were measured in a 96-well plate using a microplate reader (Bio-Rad, Hercules, CA, USA).

Specifically, the DPPH radical scavenging activity of the extracts was determined using the method of Brand-Williams, Cuvelier and Berset [32]. Briefly, 0.1 mM DPPH solution was prepared using DPPH and 80% methanol followed by a normalization at an absorbance of 0.650 ± 0.020 at 517 nm. Extract (0.05 mL) was added to 2.95 mL of the DPPH solution and reacted at room temperature for 30 min. The absorbance at 517 nm subtracted from a blank was determined. 

The ABTS radical scavenging activity of the extracts was determined by the method of Van den Berg et al. [33]. A phosphate buffer saline solution of AAPH (1.0 mM) was reacted with ABTS (2.5 mM) for 30 min at 70 °C. Subsequently, the reaction solution was filtrated through a 0.45 µm syringe filter followed by an adjustment of an absorbance to 0.650 ± 0.020 at 517 nm. ABTS reagent (980 µL) was applied to 20 µL of extract and reacted at 37 °C for 10 min. The absorbance at 517 nm subtracted from a blank was determined. 

A FRAP assay was performed according to the method of Benzie and Strain [34]. Briefly, 300 mM acetate buffer (pH 3.6), made with 3.1 g C_2_H_3_NaO_2_∙3H_2_O with 16 mL C_2_H_4_O_2_ and 10 mM TPTZ (2,4,6- tripyridyl-s-triazine) solution in 40 mM HCl, and 20 mM FeCl_3_∙6H_2_O solution was prepared and stocked. A fresh working solution of acetate buffer (25 mL), TPTZ solution (2.5 mL), and FeCl_3_∙6H_2_O solution (2.5 mL) was stored at 37 °C until application of 2850 mL FRAP solution to 150 mL of extract. The mixture was reacted for 30 min in a dark condition, and the colorimetric changes by the ferrous tripyridyltriazine complex was determined at 593 nm.

### 2.6. Total Polyphenol Content (TPC)

Total polyphenol content (TPC) of all turmeric extracts (ground turmeric, HHPE turmeric, puffed turmeric, puffed and HHPE turmeric) was determined using Folin and Ciocalteu’s assay [35]. The extract (200 µL), distilled water (2.6 mL), and Folin-Ciocalteu solution (200 µL) were left to react for 6 min. After 6 min, 2 mL of Na_2_CO_3_ was added and absorbance was measured at 750 nm after another 90 min. Gallic acid was employed as the standard and results were expressed as mg gallic acid equivalents (GAE)/g dried turmeric.

### 2.7. Quantitative Analysis of Bioactive Compounds by HPLC

Bioactive compounds in the extracts, i.e., curcumin (CUR), demethoxycurcumin (DMC), bis-demethoxycurcumin (BDMD), ferulic acid (FA), 4-vinyl guaiacol (4VG), vanillin (VN), and vanillic aicd (VA), were quantitatively determined by using an HPLC method [36]. In brief, 10 µL of extract, filtered through a 0.45 µm Millipore filter, was analyzed using the Agilent 1260 Infinity II HPLC system (Agilent Technologies, Santa Clara, CA, USA) equipped with a Zorbax SB-C18 column (4.6 × 250 mm, 5 µm, Agilent Technologies, Santa Clara, CA, USA), monitored at 260 nm. The mobile phase was run at 1 mL/min with the following differential gradient of 0.4% acetic acid in deionized water (A) and acetonitrile (B): 0–30 min, 92–9% A; 30–39 min, 9–0% A; 39–44 min, 0% A; 44–47 min, 0–92% A; 47–50 min, 92% A. The standard molecules of the curcuminoids, as well as their degradation products, were obtained from Sigma-Aldrich Co. (St. Louis, MO, USA), whose calibration curves served for the identification and quantification of the bioactive molecules in the extracts.

### 2.8. Statistical Analysis

Data are representative of three repeated experiments with three replicates. All experimental data were analyzed using one-way analysis of variance (ANOVA) and are expressed as the mean ± standard deviation (SD). Duncan’s multiple range test was conducted to assess significant differences among experimental mean values for extraction yield, antioxidant capacity, and total phenolic content using SAS software (version 8.2, SAS Institute, Inc., Cary, NC, USA). Tukey’s post-hoc multiple comparisons test was applied to the HPLC analyses using GraphPad Prism software (version 5, La Jolla, CA, USA). For all statistical analyses, (*p* < 0.05) was considered statistically significant.

## 3. Results 

### 3.1. High Hydrostatic Pressure Extraction (HHPE) Yield

The effects of various HHPE parameters, i.e., pressure, time, and ethanol concentration, on extraction yield of ground turmeric are shown in (Table 1). Pressure did not have a positive impact on extraction yield in this case. Moreover, a slight decrease of extraction yield was observed with increasing pressure level. Extraction time also exhibited only a small impact on extraction yield. A slight increase in extraction yield was observed with increasing extraction time. On the other hand, ethanol concentration greatly influenced the extraction yield of the HHPE of turmeric. The highest extraction yield was observed with 0% ethanol in the solution, while a 95% ethanol solution exhibited about an 87% reduced extraction yield. It has been reported that extraction yield was greatly influenced by ethanol concentration in puffed ginseng [29]. Those authors reported that extraction yield decreased with increasing ethanol concentration because water soluble polymers are efficiently extracted in polar solvent conditions. 

### 3.2. Factors Affecting Antioxidant Activity and Total Phenolic Content of HHPE of Ground Turmeric

#### 3.2.1. Effect of Pressure

The effects of HHPE variables, i.e., pressure, extraction time, and ethanol concentration, on antioxidant activity of the extracts were investigated (Figure 1). Pressure demonstrated very minor effects (Figure 1A): Only a slight increment of antioxidant activities of turmeric was observed in extracts at 400 MPa (DPPH, ABTS, and FRAP values at 7.35 ± 0.30, 13.82 ± 0.82, and 7.61 ± 0.21 mg VCE/g dried turmeric, respectively) as compared to 0.1 MPa atmospheric extraction (6.61 ± 0.26, 11.76 ± 1.01, and 7.17 ± 0.23 mg VCE/g dried turmeric, respectively) (Figure 1A). Total phenolic content (TPC) was not affected by the pressure level.

#### 3.2.2. Effect of Extraction Time

Antioxidant activities and TPC were analyzed in the HHPE at various extraction times (5−30 min) at 400 MPa using a 70% ethanol solvent. Following a 15 min extraction, DPPH, ABTS, and FRAP significantly increased to 7.35 ± 0.30, 13.82 ± 0.82, and 7.61 ± 0.21 mg VCE/g dried turmeric, respectively, as compared to a 5 min extraction (6.97 ± 0.52, 12.80 ± 0.73, and 7.27 ± 0.37 mg VCE/g dried turmeric, respectively). Further extension of extraction time reached the maximal increment of all antioxidant activities in the HHPE turmeric at 20 min (*p* < 0.05) (Figure 1B). TPC also significantly increased to its maximum in a 20 min as compared to a 5 min extraction (6.12 ± 0.10 vs. 5.62 ± 0.49 mg gallic acid equivalents (GAE)/g dried turmeric, respectively) (*p* < 0.05).

#### 3.2.3. Effect of Ethanol Concentration

Ethanol concentration greatly influenced the antioxidant activity of HHPE turmeric (Figure 1C). All antioxidant activities increased with increasing ethanol concentration up to 70% (*p* < 0.05) at which the maximal effects were observed (DPPH 7.45 ± 0.17, ABTS 13.85 ± 0.49, and FRAP 7.77 ± 0.13 mg VCE/g dried turmeric, respectively). TPC also significantly increased with increasing ethanol concentration giving 6.57 ± 0.37 mg GAE/g dried turmeric in 95% ethanol up from 1.19 ± 0.11 mg GAE/g dried turmeric in 0% ethanol water. 

### 3.3. HPLC Analysis of HHPE of Ground Turmeric

The bioactive compounds in the HHPE turmeric were analyzed using HPLC (Table 2). Curcumin (CUR), demethoxycurcumin (DMC), and bis-demethoxycurcumin (BDMC) were determined to be the major curcuminoids in turmeric extract, and their degradation products, ferulic acid (FA), vanillin (VN), and vanillic acid (VA) [37], were also found in the HHPE turmeric. Pressure level did not greatly influence the amount of the bioactive compounds of HHPE turmeric even though a slight but significant decrement of FA and VN was observed (Table 2). In contrast, extraction time significantly influenced the composition of bioactive compounds in the HHPE turmeric. The amount of curcuminoids decreased and the amount of the minor bioactive compounds (VA and FA) increased with increasing extraction time. Ethanol concentration also greatly influenced the composition of bioactive compounds in the HHPE turmeric. CUR content dramatically increased with increasing ethanol concentration from 5.31 (0% ethanol) to 407.65 µg/ g dried turmeric (70% ethanol). Additionally, DMC and BDMC were not detected in up to 20% and 40% ethanol concentrations, respectively. 

### 3.4. Combinatory Effect of Puffing and HHPE on Turmeric

Following the investigation on the optimal conditions for HHPE turmeric, which were determined to be at 400 MPa for 20 min using 70% ethanol, the combinatory effect of puffing at 980 kPa and HHPE on turmeric was determined by assessing extraction yield, antioxidant activity, TPC, and the bioactive compounds profile. Neither puffing nor HHPE alone showed any effect in the extraction yield of puffed and HHPE turmeric (Table 3). Puffing of turmeric followed by HHPE, however, demonstrated significantly increased extraction yield to 11.53% as compared to non-puffed turmeric extracted at atmospheric pressure (0.1 MPa) (*p* < 0.05).

On the other hand, puffing and HHPE did not show any synergistic effect on antioxidant activity and TPC (Figure 2). Puffing showed the highest antioxidant activity (11.89, 21.24, 12.98 mg VCE/g dried turmeric in DPPH, ABTS, and FRAP, respectively) and TPC at 13.49 mg GAE/g dried turmeric, and puffing followed by HHPE showed slightly lower antioxidant activity (11.58, 21.04, 12.75 mg VCE/g dried turmeric in DPPH, ABTS, FRAP, respectively) and TPC at 12.58 mg GAE/g dried turmeric. Consequently, the combination of puffing and HHPE is not beneficial to antioxidant activity and total phenolic content compared to the single treatment of puffing. Overall, puffing is the best way to increase the antioxidant activity and TPC of turmeric. 

The combinatory effect of puffing and HHPE on bioactive compounds of turmeric was analyzed using HPLC (Figure 3) and quantitatively analyzed (Table 4). Puffing showed an increase in minor bioactive compounds and decrease in curcuminoids. 4VG was not observed in non-puffed turmeric, but detected in all puffed turmeric and increased with HHPE. 

## 4. Discussion

The current study sought to find the best condition of HHPE for the augmentation of turmeric’s antioxidant properties. In addition, the combination of the puffing process with HHPE, for which the enhanced antioxidant capacity was clearly demonstrated [3], was investigated for clarification of any synergistic and/or additive effects.

The main parameters of turmeric HHPE, i.e., pressure, time, and ethanol concentration, were investigated. Pressure and extraction time in HHPE have a very marginal influence on the extraction yield of turmeric. In contrast, HHPE of ginseng powder exhibited improved extraction yield in a pressure and time dependent manner [29]. This discrepancy may come from the difference in soluble components between turmeric vs. ginseng powder. Of interest, ethanol concentration in HHPE greatly affected the extraction yield and the amount of major bioactive compounds (CUR, DMC, and BDMC) of turmeric. The hydrophobicity of the major curcuminoids (CUR, DMC, and BDMC) made them not well solubilized in water. Consequently, BDMC revealed the highest hydrophobicity followed by DMC and CUR. At 0% ethanol concentration the highest yield was observed, but only a very small amount of the major curcuminoids was detected, suggesting that most of the extracts from turmeric are water soluble. Most of the water-soluble components in turmeric, mainly dietary fibers, are not extracted at higher ethanol concentrations, resulting in a drastic decrease in extraction yield at a 95% ethanol concentration. Similarly, it has been reported that increased ethanol concentration negatively affected the extraction yield of red-ginseng [38] and that the high HHPE yield of ginseng was shown in a lower ethanol concentration, suggesting that water soluble molecules including starches were the main components in the extraction process [29,38].

Extraction pressure and time were also found to have very marginal to no impacts on antioxidant activity and TPC. This apparent discrepancy may demonstrate the hydrophobicity of the antioxidant components in turmeric. A significant change in bioactive compounds of turmeric in HHPE was not observed in this study. High pressure level (550 MPa) and long treatment time (30 min) had no impact on the HHPE of bioactive compounds. Consequently, the optimal conditions based on antioxidant activity for the HHPE of non-puffed turmeric were determined to be 400 MPa for 20 min using 70% ethanol. 

HHPE increased the curcuminoids and minor bioactive compounds except FA in puffed turmeric. This suggests that curcuminoids are degraded to minor bioactive compounds by puffing. At first, curcuminoids decompose to FA and then the FA degrades further to produce 4VG. Due to the differences in degradation rates of curcuminoids-to-FA and FA-to-4VG, FA may be continuously accumulated despite its decomposition to 4VG [5,37]. It has been reported that curcumin is degraded and transformed to volatile phenolic compounds such as vanillin, guaiacol, and isoeugenol [5]. Curcumin may be further degraded into ferulic acid, 4-vynilguaiacol, vanillic acid, and vanillin in thermal treatment. The degradation rate of curcumin to ferulic acid may be faster than the degradation rate of ferulic acid to 4-vynilguaiacol. Finally, degradation to volatile compounds, such as vanillic acid or vanillin, occurred slowly [37]. The HPLC analysis in the current study suggests that curcuminoids are highly insoluble in water, and BDMC has the highest insolubility to water followed by DMC and CUR. In contrast to the trend seen in curcuminoids, VA content decreased with increasing ethanol concentration, indicating that VA was more water soluble than ethanol. While previous studies showed an increased amount of bioactive compounds using HHPE for anthocyanin in grapes, carotenoids in carrot puree, and lycopene in tomato puree [39,40], no significant increase in bioactive compounds in HHPE of ground turmeric was observed in this study. Moreover, intense conditions such as high pressure level (550 MPa) and long treatment time (30 min) were not necessary for better extraction of bioactive compounds in HHPE of ground turmeric. This is in line with the results of antioxidant activities of HHPE in this study and the results of Casquete et al. [41]. Overall, other bioactive compounds except VA had the highest amounts in the HHPE using 70% ethanol. Similarly, 60% ethanol extraction for phenolic compounds in flaxseed, and 70% ethanol extraction for high antioxidant activity in ginseng have been previously recommended [29,42]. On the other hand, combination of puffing and HHPE showed an increase in extraction yield and bioactive compounds. Assessment of antioxidant capacity further revealed that puffing is an effective method to increase antioxidant capacity, but HHPE after puffing did not show any synergistic nor additive impacts in antioxidant capacity. Consequently, HHPE was found to be a more effective method than conventional extraction in the case of the porous and weak-structured materials created by puffing. 

## 5. Conclusions

The best HHPE conditions for extraction of turmeric to maximize the antioxidant activity were investigated. The synergistic effect of puffing and HHPE on antioxidant activity and bioactive compounds of turmeric was also investigated. Ethanol concentration was the most effective variable for HHPE of ground turmeric among the three studied variables of ethanol concentration, pressure level, and treatment time. HHPE of ground turmeric at 400 MPa for 20 min with 70% EtOH was the best extraction condition for the highest antioxidant activity. Extraction of bioactive compounds in the HHPE of ground turmeric was also greatly influenced by ethanol concentration, possibly due to their hydrophilic or hydrophobic characteristics. Although puffing and HHPE showed an increase in extraction yield and bioactive compounds, a synergistic effect was not observed on antioxidant activity. 

## Figures and Tables

**Figure 1 foods-09-01690-f001:**
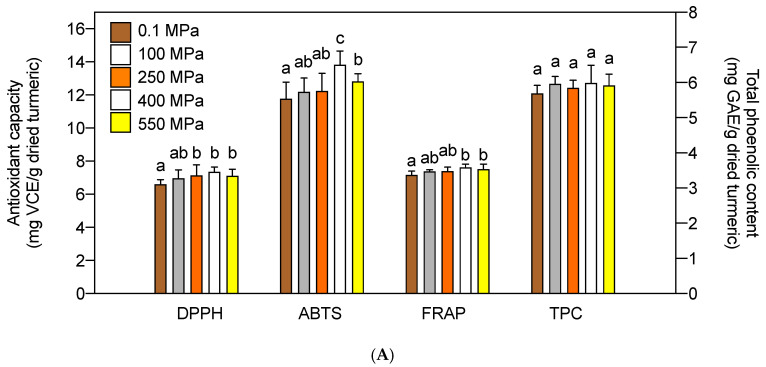
Effects of pressure (**A**), extraction time (**B**), and ethanol concentration (**C**) on antioxidant activities and total phenolic contents (TPC) of the HHPE turmeric. Same letters in each group indicates that they are not statistically different at *p* < 0.05.

**Figure 2 foods-09-01690-f002:**
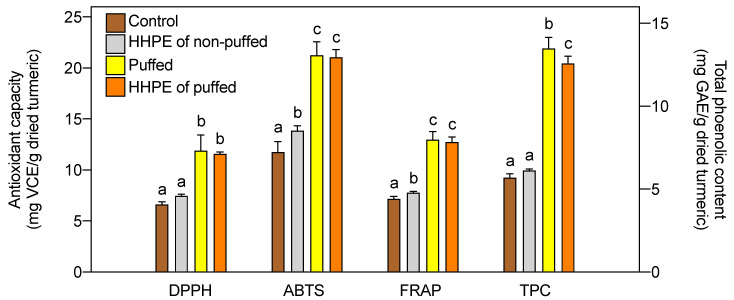
Combination effect of puffing and HHPE on antioxidant activities and total phenolic contents (TPC) of turmeric. Same letters in each group indicates that they are not statistically different at *p* < 0.05.

**Figure 3 foods-09-01690-f003:**
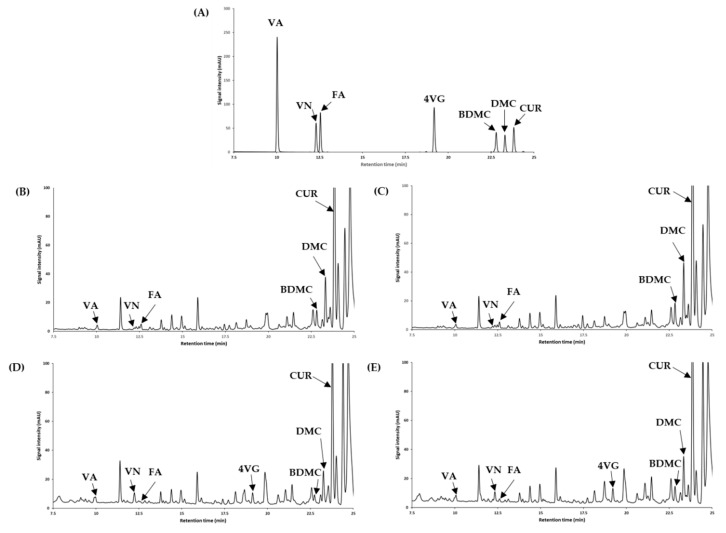
HPLC chromatograms of seven standard materials including curcuminoids and their degradation products (**A**), turmeric extract (**B**), HHPE turmeric (**C**), puffed turmeric (**D**), and puffed and HHPE turmeric (**E**) for quantification. VA, vanillic acid; VN, vanillin; FA, ferulic acid; 4VG, 4-vinyl guaiacol; BDMC, bisdemethoxycurcumin; DMC, demethoxycurcumin; CUR, curcumin.

**Table 1 foods-09-01690-t001:** High hydrostatic pressure extraction (HHPE) yield of ground turmeric.

Pressure (MPa)	Extraction Yield (%)	Time (min)	Extraction Yield (%)	Ethanol Concentration (%)	Extraction Yield (%)
0.1	10.77 ± 0.24 ^a,^*	5	8.46 ± 0.38 ^b^	0	20.04 ± 0.82 ^a^
100	10.60 ± 0.30 ^a^	10	9.99 ± 0.54 ^a^	20	17.24 ± 0.37 ^b^
250	10.55 ± 0.27 ^a^	15	10.23 ± 0.24 ^a^	40	16.85 ± 0.40 ^b^
400	10.23 ± 0.24 ^b^	20	10.60 ± 0.71 ^a^	70	10.60 ± 0.71 ^d^
550	10.12 ± 0.22 ^b^	30	10.77 ± 0.75 ^a^	95	2.52 ± 0.01 ^e^

* Values with the same letters in the same column are not significantly different (*p* < 0.05).

**Table 2 foods-09-01690-t002:** Effects of pressure, extraction time, and ethanol concentration on bioactive compounds (µg/g dried turmeric) in the HHPE turmeric analyzed by HPLC.

Pressure	VA **	VN	FA	4VG	BDMC	DMC	CUR
**0.1 MPa**	3.46 ± 0.32 ^a,^*	32.12 ± 2.85 ^a^	20.46 ± 0.44 ^a^	N/D ***	23.29 ± 0.12 ^b,c^	100.91 ± 1.72 ^a^	399.96 ± 2.49 ^a^
**100 MPa**	2.67 ± 0.40 ^b,c^	28.96 ± 0.89 ^b^	19.05 ± 0.21 ^a,b^	N/D	21.87 ± 0.22 ^c^	95.44 ± 3.19 ^a^	395.65 ± 5.07 ^a^
**250 MPa**	2.40 ± 0.28 ^b,c^	28.16 ± 0.46 ^b^	18.50 ± 0.18 ^b^	N/D	20.85 ± 0.58 ^d^	98.24 ± 1.56 ^a^	396.81 ± 5.39 ^a^
**400 MPa**	2.23 ± 0.18 ^c^	19.87 ± 1.04 ^c^	16.26 ± 0.52 ^c^	N/D	24.70 ± 0.25 ^a,b^	95.65 ± 2.64 ^a^	390.72 ± 1.97 ^a^
**550 MPa**	3.04 ± 0.45 ^a,b^	17.79 ± 0.30 ^c^	14.52 ± 0.38 ^d^	N/D	25.89 ± 0.28 ^a^	95.97 ± 0.75 ^a^	399.57 ± 3.79 ^a^
**Extraction Time**							
**5 min**	1.86 ± 0.00 ^d^	13.71 ± 0.11 ^d^	21.82 ± 1.05 ^a^	N/D	29.37 ± 0.04 ^a^	126.6 ± 0.21 ^a^	438.83 ± 1.14 ^a^
**10 min**	2.94 ± 0.21 ^b,c^	13.64 ± 0.77 ^d^	20.27 ± 2.55 ^b^	N/D	25.73 ± 0.75 ^a^	115.7 ± 2.60 ^b^	400.32 ± 8.23 ^b,c^
**15 min**	2.23 ± 0.18 ^c,d^	19.87 ± 1.04 ^c^	16.26 ± 0.52 ^c^	N/D	24.70 ± 0.25 ^a^	95.65 ± 2.64 ^c^	390.72 ± 1.97 ^b,c^
**20 min**	3.51 ± 0.65 ^a,b^	28.02 ± 2.36 ^b^	20.36 ± 1.04 ^b^	N/D	24.40 ± 2.35 ^a^	100.1 ± 2.03 ^c^	407.65 ± 2.63 ^b^
**30 min**	4.20 ± 0.93 ^a^	30.92 ± 1.06 ^a^	19.47 ± 1.01 ^b^	N/D	23.66 ± 2.49 ^a^	100.9 ± 1.21 ^c^	408.51 ± 3.26 ^b^
**EtOH Concentration**							
**0%**	7.23 ± 0.43 ^a^	19.49 ± 2.69 ^b^	N/D	N/D	N/D	N/D	5.31 ± 0.06 ^d^
**20%**	4.31 ± 0.27 ^b^	20.23 ± 1.43 ^b^	N/D	N/D	N/D	N/D	9.65 ± 0.05 ^d^
**40%**	4.37 ± 1.40 ^b^	20.96 ± 0.84 ^b^	1.25 ± 0.14 ^c^	N/D	N/D	4.12 ± 0.11 ^b^	110.74 ± 3.98 ^c^
**70%**	3.51 ± 0.65 ^c^	28.02 ± 2.36 ^a^	20.36 ± 1.04 ^a^	N/D	24.40 ± 2.35 ^a^	100.12 ± 2.03 ^a^	407.65 ± 2.63 ^a^
**95%**	N/D	20.64 ± 0.10 ^b^	26.96 ± 0.68 ^a^	N/D	22.29 ± 0.38 ^a^	94.50 ± 1.44 ^a^	389.67 ± 5.57 ^a^

* Values designated by different letters (a–d) are statistically different at *p* < 0.05 within the column. ** VA, vanillic acid; VN, vanillin; FA, ferulic acid; 4VG, 4–vinyl guaiacol; BDMC, bisdemethoxycurcumin; DMC, demethoxycurcumin; CUR, curcumin. *** N/D, not detected.

**Table 3 foods-09-01690-t003:** Effect of the combination of puffing and HHPE on extraction yield of turmeric.

Treatment	Extraction Yield (%)
Control	10.77 ± 0.24 ^b,^*
HHPE	10.60 ± 0.71 ^b^
Puffing	10.84 ± 0.94 ^b^
Puffing + HHPE	11.53 ± 0.73 ^a^

* Values designated by different letters (a,b) are statistically different at *p* < 0.05 within the column.

**Table 4 foods-09-01690-t004:** Effect of the combination of puffing and HHPE on bioactive compounds (g/g dried turmeric) of turmeric.

Treatment	VA **	VN	FA	4VG	BDMC	DMC	CUR
Control	3.46 ± 0.32 ^c,^*	32.12 ± 2.85 ^c^	20.46 ± 0.44 ^a^	N/D ***	23.29 ± 0.12 ^a^	100.91 ± 1.72 ^a^	399.96 ± 2.49 ^b^
HHPE	3.51 ± 0.65 ^c^	28.02 ± 2.36 ^d^	20.36 ± 1.04 ^a^	N/D	24.40 ± 2.35 ^a^	100.12 ± 2.03 ^a^	407.65 ± 2.63 ^a^
Puffing	13.66 ± 0.83 ^b^	79.22 ± 6.00 ^a^	6.07 ± 1.29 ^b^	77.61 ± 5.30 ^b^	10.46 ± 1.32 ^c^	63.58 ± 3.80 ^c^	306.53 ± 18.96 ^d^
Puffing + HHPE	16.25 ± 1.16 ^a^	72.55 ± 1.99 ^b^	5.78 ± 0.42 ^b^	140.00 ± 3.14 ^a^	16.54 ± 1.10 ^b^	70.04 ± 1.29 ^b^	352.41 ± 3.47 ^c^

* Values designated by different letters (a–d) are statistically different at *p* < 0.05 within the col umn. ** VA, vanillic acid; VN, vanillin; FA, ferulic acid; 4VG, 4–vinyl guaiacol; BDMC, bisdemethoxycurcumin; DMC, demethoxycurcumin; CUR, curcumin. *** N/D, not detected.

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
