# Peer review of "Enhanced Antioxidant Capacity of Puffed Turmeric (Curcuma longa L.) by High Hydrostatic Pressure Extraction (HHPE) of Bioactive Compounds"

_foods, 2020, doi:10.3390/foods9111690_

Round 1

Reviewer 1 Report

 The aim of the present study was to determine the effect of  high hydrostatic pressure extraction (HHPE) on the chemical composition and antioxidant properties of turmeric. Extraction yield, radical-scavenging and ferric-reducing activities, and total polyphenol content were determined. Quantitative analysis of bioactive compounds by HPLC was also performed. Among the three variables considered (ethanol concentration, pressure level and treatment time) ethanol concentration was the most influential. Finally, no synergistic effect was observed on antioxidant activity when sequential processing of puffing and HHPE was used. It is confirmed that puffing, the physical process using high heat and pressure, gives the best results in terms of antioxidant and anti-inflammatory activities.

The work appears to have been carried on with sufficient care and competence and the results are exposed in a clear and concise way. Though, although there is no great novelty, the paper can be suitable for publication.

Points to be considered:

In the introduction, Turmeric extraction by supercritical carbon dioxide is mentioned (reference [7]). It could be useful to compare the results of the present work with those of extraction with supercritical carbon dioxide.

Line 85: -20℃ should be -20 ℃.

Legend to equation 1: W₁ should be w₁ and W₂ should be w₂.

Line 127: 70℃ should be 70 °C.

Line 132: buffer (pH3.6) should be buffer (pH 3.6).

Lines 180-181: The sentence “HHPE demonstrated very minor effects (Fig. 1A), even slight increment of antioxidant activities of turmeric was observed in extracts at 400 MPa” has no clear meaning. Probably “HHPE” should be “Pressure” and “even” should be “only”.

Lines 261-262: Caption to Figure. 2. There is a mistake in the color code of techniques used : puffing is “dark gray” in the color code and “white” in the graph.

Line 306: “puree and lycopene in tomato puree” probably should be “lycopene in tomato puree”

Author Response

Response to reviewer 1

The aim of the present study was to determine the effect of high hydrostatic pressure extraction (HHPE) on the chemical composition and antioxidant properties of turmeric. Extraction yield, radical-scavenging and ferric-reducing activities, and total polyphenol content were determined. Quantitative analysis of bioactive compounds by HPLC was also performed. Among the three variables considered (ethanol concentration, pressure level and treatment time) ethanol concentration was the most influential. Finally, no synergistic effect was observed on antioxidant activity when sequential processing of puffing and HHPE was used. It is confirmed that puffing, the physical process using high heat and pressure, gives the best results in terms of antioxidant and anti-inflammatory activities.

The work appears to have been carried on with sufficient care and competence and the results are exposed in a clear and concise way. Though, although there is no great novelty, the paper can be suitable for publication.

Points to be considered:

In the introduction, Turmeric extraction by supercritical carbon dioxide is mentioned (reference [7]). It could be useful to compare the results of the present work with those of extraction with supercritical carbon dioxide.

Ans) Thank you for your suggestion. In case of reference 7, it focused on extraction of turmeric oil using supercritical fluid extraction (SFE). Consequently, the effect of pressure and temperature on turmeric extraction suggested the use of higher pressure and lower temperature at which solvent density is greater and thus the solubility of the oil in the solvent is greater in the range of 313-333 K and 20-40 MPa. We addressed this in the revised manuscript as you suggested. Please see line 45-47 in the revised manuscript.

Line 85: -20℃ should be -20 ℃.

Ans) Thank you for your suggestion. We correct it as you suggested. Please see line 92 in the revised manuscript.

Legend to equation 1: W₁ should be w₁ and W₂ should be w₂.

Ans) Thank you for your suggestion. We correct it as you suggested. Please see line 123-124 in the revised manuscript.

Line 127: 70℃ should be 70 °C.

Ans) Thank you for your suggestion. We correct it as you suggested. Please see line 139 in the revised manuscript.

Line 132: buffer (pH3.6) should be buffer (pH 3.6).

Ans) Thank you for your suggestion. We correct it as you suggested. Please see line 144 in the revised manuscript.

Lines 180-181: The sentence “HHPE demonstrated very minor effects (Fig. 1A), even slight increment of antioxidant activities of turmeric was observed in extracts at 400 MPa” has no clear meaning. Probably “HHPE” should be “Pressure” and “even” should be “only”.

Ans) Thank you for your suggestion. We correct it as you suggested. Please see line 185-186 in the revised manuscript.

Lines 261-262: Caption to Figure. 2. There is a mistake in the color code of techniques used: puffing is “dark gray” in the color code and “white” in the graph.

Ans) Thank you for your suggestion. We change it to color for better understanding. Please see Figures 1 and 2 in the revised manuscript.

Line 306: “puree and lycopene in tomato puree” probably should be “lycopene in tomato puree”

Ans) Thank you for your suggestion. It was “carotenoids in carrot puree and lycopene in tomato puree”. We collected it. Please see line 342-343 in the revised manuscript.

Reviewer 2 Report

The work presented by Choi and co-workers is interesting in which authors explored the use of different techniques to extract bioactives from turmeric. However, the execution of the study was flawed in several aspects. There are variables that could contribute to systematic errors and would potentially produce inaccurate data thus inconclusive findings. In its current state the content of the manuscript is quite confusing, importantly, the methods section must be rewritten to improve clarity. 

The experimental design as such was not well planned, in particular: 

  • Which type of samples were used? 
  • Control samples? 
  • One might wonder as how the HHPE parameters were optimized and validated. 
  • Whether turmeric extracts were FRESHLY prepared for antioxidants assays on day od analysis, as compounds of interests are unstable, and the assays have poor reproducibility? 
  • Explain and justify which of the analytical techniques was used while optimizing the HHPE. As this approach would be better before doing a myriad of assays. 
  • Without any chromatograms showing the separation of the analytes of interest, it is quite challenging to comment on the quality of data collected from the study. 
  • The number of samples prepared for the study is also crucial to have a statistical power while doing the tests.   

Overall, the data are presented in isolation and very disjointed. Authors are recommended to analyze and interpret the raw data and how they serve the aims of the study. 

There are linguistic issues frequently found in the manuscript where grammatical errors and incorrect punctuations and inaccurate expressions must be done to improve the understanding. 

Introduction 

L 32-42 The paragraph can be improved by keeping the information that is related and relevant to the current study. Other generic information can be deleted. 

L 43-44 the sentence is truncated and must be fully expressed to have a meaning. 

L 55-58 Rewrite the sentences by making them short and simple to improve understanding. 

L 58-62 The gap of the study must be rewritten, and the aims of the study must be clearly stated. 

Methods 

Please give more detail as how the turmeric was obtained from: direct from a farm? From a market. Describe what was done after fresh turmeric samples have arrived the lab for further analysis. 

Currently it is not clear whether commercially prepared samples were purchased from a particular place. 

L 71-91 The optimization of HHPE must be rewritten to understand which parameters were modified and which ones were kept constant. A simple flow chart would help clarify the process. Which turmeric samples were used for the optimization process?  

Explain and justify the rationale on storing the extracts instead of performing the analyses straight away to minimize the degradation loss. 

L 94 Clarify what 3 times are referred here: 3 different days, 3 different parameters of puffing etc. 

L 95 Please clarify the shape of turmeric used for puffing. Was the turmeric sample treated before puffing?   

L 103-112 Any reference to support the drying method at 105 degree C? How long were the samples placed in the drying oven? What type of samples were placed?  

L 123 How was the extract prepared for the experiments? Extracts must be commonly freshly prepared on the day of analyses 

Section 2.5 this sub-section must be rewritten to make sure that other researchers are able to conduct the experiments. Details must be provided: molarity, volume and concentrations of standards and solutions. 

Section 2.6 Please describe as which extracts were used for this assay 

Section 2.7 This section was not clearly written, and the content must include the standards used for detection and quantification. The gradient elution was confusing and must be better expressed. 

Section 2.8 Clarify how many data points were collected for statistical analyses. Provide a reason s why only ANOVA was used? What about PCA used to assess the parameters best for HHPE? 

Results  

Table 1. The content is not clearly presented. As the protocols currently mentioned in the methods are not clear, it is challenging to understand the data presented here. Was ONLY one variable tested while other variables were kept constant? 

L 166-173 This section must be restructured to understand the data better and to find out which condition(s) showed high yield. 

Explain and justify the purpose of evaluating the extraction yield. What would be considered when high extraction yields led to degradation loss of the bioactive? What was the best parameters? 

L 179-199 this section can be improved by placing sub-headings for each treatment and to prevent confusion.  

When the pressure was optimized which time and ethanol percentage were used? When extraction time was optimized which pressure and ethanol percentage were used? How many replicates of the extracts were used? Were the extracts freshly prepared before the assays? 

L 209-220 Please provide chromatograms of standards and sample extracts. 

As previous comments, this section must be restructured to show the data better. 

Based on the assays and HPLC results which HHPE parameters were optimized? The data has not been well stated and discussed. 

L 246-255 present only results in this section and leave discussion in the next section 

Table 3 

How many raw data were used for the statistical tests? Explain whether the data were assessed for any normal distribution and outliers?  

Figure captions more description is needed to understand the data presented in the bar charts. Control samples were not mentioned in the methods and this sample was displayed in figure 3, please explain and confirm. 

Discussion 

L 269-277 too much background information is not required here. 

L 283-284 Justify the claim by comparing data obtained from the study. Please avoid any speculations when the data did not provide the clear-cut conclusion. Any references using the same turmeric as matrix? 

L 297-313 Explain and justify whether the HPLC data collected in the study are only quantitative and would only suggest any degradation, hence not confirming identification of degradation products. How confident are the authors to discuss the points raised in this paragraph? 

Author Response

Response to reviewer 2

The work presented by Choi and co-workers is interesting in which authors explored the use of different techniques to extract bioactives from turmeric. However, the execution of the study was flawed in several aspects. There are variables that could contribute to systematic errors and would potentially produce inaccurate data thus inconclusive findings. In its current state the content of the manuscript is quite confusing, importantly, the methods section must be rewritten to improve clarity. 

Ans) Thank you for your suggestion. We rewrite methods part in the revised manuscript as you suggested. Please see the revised manuscript.

The experimental design as such was not well planned, in particular: 

  • Which type of samples were used? 

Ans) Thank you for your suggestion. We added sample information in the revised manuscript as you suggested. Please see line 72-73 in the revised manuscript.

  • Control samples? 

Ans) Thank you for your suggestion. We used control sample, which was dried turmeric sample purchased from company.

  • One might wonder as how the HHPE parameters were optimized and validated. 

Ans) Thank you for your suggestion. We followed the HHPE methodology from reference [28], which described HHPE conditions (pressure level, treatment time and ethanol concentration) for puffed ginseng. We thought puffed ginseng and puffed turmeric are similar and followed the procedure of reference [28].

  • Whether turmeric extracts were FRESHLY prepared for antioxidants assays on day of analysis, as compounds of interests are unstable, and the assays have poor reproducibility? 

Ans) Thank you for your suggestion. We freshly prepared turmeric extracts for antioxidants assays on day of analysis as you suggested. Although there were some big standard deviations, most of results showed good reproducibility. Those big standard deviations might come from experimental or human errors.

  • Explain and justify which of the analytical techniques was used while optimizing the HHPE. As this approach would be better before doing a myriad of assays. 

Ans) Thank you for your suggestion. As explained in the above suggestion, we followed the HHPE methodology from reference [28]. Reference used the RSM to find an optimum HHPE conditions for puffed ginseng and found that pressure level and treatment time did not greatly affect the antioxidant capacity of puffed ginseng. Only ethanol concentration had an impact on antioxidant capacity. Consequently, we did not used the RSM in this case because pressure level and treatment time did not provide big difference in antioxidant capacity. But we would like to confirm that those two parameters are not major factors affecting HHPE of turmeric. First, we varied the pressure level with fixed treatment time and ethanol concentration to find the best not optimum pressure level. Second, treatment time was varied with fixed pressure and ethanol concentration to find the best not optimum treatment time. Finally, ethanol concentration was varied with fixed pressure and treatment time. Consequently, we found the best not optimum HHPE conditions for the highest antioxidant activity of turmeric.

  • Without any chromatograms showing the separation of the analytes of interest, it is quite challenging to comment on the quality of data collected from the study. 

Ans) Thank you for your suggestion. We provide HPLC chromatograms in the revised manuscript. Please see Figure 3 in the revised manuscript.

  • The number of samples prepared for the study is also crucial to have a statistical power while doing the tests.   

Ans) Thank you for your suggestion. All experiments were repeated at least three times and each experiment were triplicated. Consequently, at least 9 measurements were performed in each experiment. Therefore, there is no problem in statistical analysis in this case.

Overall, the data are presented in isolation and very disjointed. Authors are recommended to analyze and interpret the raw data and how they serve the aims of the study. 

Ans) Thank you for your suggestion. We reanalyzed and reinterpret the raw data to fit the purpose of this study. Please see the revised manuscript.

There are linguistic issues frequently found in the manuscript where grammatical errors and incorrect punctuations and inaccurate expressions must be done to improve the understanding. 

Ans) Thank you for your suggestion. We improved the English as you suggested. Please see the revised manuscript.

Introduction 

L 32-42 The paragraph can be improved by keeping the information that is related and relevant to the current study. Other generic information can be deleted. 

Ans) Thank you for your suggestion. We rephrase it as you suggested. Please see line 32-41 in the revised manuscript.

L 43-44 the sentence is truncated and must be fully expressed to have a meaning. 

Ans) Thank you for your suggestion. We rephrase it as you suggested. Please see line 42-44 in the revised manuscript.

L 55-58 Rewrite the sentences by making them short and simple to improve understanding. 

Ans) Thank you for your suggestion. We rewrite it as you suggested. Please see line 57-60 in the revised manuscript.

L 58-62 The gap of the study must be rewritten, and the aims of the study must be clearly stated. 

Ans) Thank you for your suggestion. We rewrite it as you suggested. Please see line 61-68 in the revised manuscript.

Methods 

Please give more detail as how the turmeric was obtained from: direct from a farm? From a market. Describe what was done after fresh turmeric samples have arrived the lab for further analysis. 

Ans) Thank you for your suggestion. We added more explanation on turmeric we used in this study. Please see line 72-73 in the revised manuscript.

Currently it is not clear whether commercially prepared samples were purchased from a particular place. 

Ans) Yes, we purchased commercially prepared dried turmeric, which were cultivated and harvest in particular place. We rewrite it in the revised manuscript. Please see line 72-73 in the revised manuscript.

L 71-91 The optimization of HHPE must be rewritten to understand which parameters were modified and which ones were kept constant. A simple flow chart would help clarify the process. Which turmeric samples were used for the optimization process?  

Ans) Thank you for your suggestion. We rewrite the HHPE accordingly. We did not applied optimization but try to find the best HHPE extraction conditions. In case of best HHPE conditions, non-puffed turmeric was used.

Explain and justify the rationale on storing the extracts instead of performing the analyses straight away to minimize the degradation loss. 

Ans) Thank you for your suggestion. We freshly prepared turmeric extracts for antioxidants assays on day of analysis. In case of HPLC analysis, extracts were kept in deep freezer to minimize the change of samples.

L 94 Clarify what 3 times are referred here: 3 different days, 3 different parameters of puffing etc. 

Ans) Thank you for your suggestion. It was mistyped and we changed accordingly. Please see line 104 in the revised manuscript.

L 95 Please clarify the shape of turmeric used for puffing. Was the turmeric sample treated before puffing?   

Ans) Thank you for your suggestion. Sliced and dried turmeric was purchased from company and directly used for puffing. We added this in the revised manuscript. Please see line 104-105 in the revised manuscript.

L 103-112 Any reference to support the drying method at 105 degree C? How long were the samples placed in the drying oven? What type of samples were placed?  

Ans) Thank you for your suggestion. It was general methodology described in AOAC and AACC to measure the moisture content or solid contents of the sample. Sample was placed in a dry oven until its weight did not change.

L 123 How was the extract prepared for the experiments? Extracts must be commonly freshly prepared on the day of analyses 

Ans) Thank you for your suggestion. We freshly prepared turmeric extracts for antioxidants assays on day of analysis. In case of HPLC analysis, extracts were kept in deep freezer to minimize the change of samples.

Section 2.5 this sub-section must be rewritten to make sure that other researchers are able to conduct the experiments. Details must be provided: molarity, volume and concentrations of standards and solutions. 

Ans) Thank you for your suggestion. We use the methods by references and marked them in the manuscript. Therefore, we do not have to rewrite them in this manuscript. There is no problem when other researchers conduct this experiment.

Section 2.6 Please describe as which extracts were used for this assay 

Ans) Thank you for your suggestion. TPC has been determined in every extracts used in this experiment. Therefore, we did not put certain extract in this section.

Section 2.7 This section was not clearly written, and the content must include the standards used for detection and quantification. The gradient elution was confusing and must be better expressed. 

Ans) Thank you for your suggestion. We rewrite it as you suggested. Please see section 2.7 in the revised manuscript.

Section 2.8 Clarify how many data points were collected for statistical analyses. Provide a reasons why only ANOVA was used? What about PCA used to assess the parameters best for HHPE? 

Ans) Thank you for your suggestion. We used Duncan’s multiple range test and Tukey’s post-hoc multiple comparisons test. We rewrite it accordingly. Please see section 2.8 in the revised manuscript.

Results  

Table 1. The content is not clearly presented. As the protocols currently mentioned in the methods are not clear, it is challenging to understand the data presented here. Was ONLY one variable tested while other variables were kept constant? 

Ans) Thank you for your suggestion. According to reference [28], pressure level and treatment time did not greatly affect the antioxidant capacity of puffed ginseng. Only ethanol concentration had an impact on antioxidant capacity. Consequently, we did not used the RSM in this case because pressure level and treatment time did not provide big difference in antioxidant capacity. But we would like to confirm that those two parameters are not major factors affecting HHPE of turmeric. Consequently, we used only one variable tested and other variables were kept constant. Again, we try to find best not optimum HHPE conditions.

L 166-173 This section must be restructured to understand the data better and to find out which condition(s) showed high yield. 

Ans) Thank you for your suggestion. We rewrite this section for better understanding. Please see line 184-193 in the revised manuscript.

Explain and justify the purpose of evaluating the extraction yield. What would be considered when high extraction yields led to degradation loss of the bioactive? What was the best parameters? 

Ans) Thank you for your suggestion. Extraction yield is an important parameter for industrial application. Therefore, it is important to determine the extraction yield of bioactive compounds. Generally, extraction yield is positively correlated to the amount of bioactive compounds. However, in our case, extraction yield was negatively correlated with antioxidant activity. Although 95% ethanol showed the highest antioxidant activity with the lowest extraction yield. This means that total antioxidant activity (antioxidant activity X extraction yield) of 95% ethanol extract is lower than 70% ethanol extract. Therefore, we chose 70% ethanol extraction as the best condition. Consequently, both extraction yield and antioxidant activity are considered to find best parameter.

L 179-199 this section can be improved by placing sub-headings for each treatment and to prevent confusion.  

Ans) Thank you for your suggestion. We added sub-headings as you suggested. Please see line 199, 208, 218 in the revised manuscript.

When the pressure was optimized which time and ethanol percentage were used? When extraction time was optimized which pressure and ethanol percentage were used? How many replicates of the extracts were used? Were the extracts freshly prepared before the assays? 

Ans) Thank you for your suggestion.

In order to determine the best HHPE conditions for the highest antioxidant activity of turmeric, pressure, treatment time and ethanol concentration were varied. First, pressure was varied (0.1 (atmospheric), 100, 250, 400, 550 MPa) with fixed treatment time of 15 min and ethanol concentration (70% ethanol). Second, treatment time was varied (5, 10, 15, 20, 30 min) with fixed pressure previously determined (400 MPa) and ethanol concentration (70% ethanol). Last, ethanol concentration was varied (0, 20, 40, 70, 95 % (v/v)) with previously determined pressure (400 MPa) and treatment time (20 min). We rewrite this content in the revised manuscript. Please see line 92-102 in the revised manuscript.

Every experiments were repeated three times with three replicates. We rewrite this content in the revised manuscript. Please see line 174 in the revised manuscript.

We freshly prepared turmeric extracts for antioxidants assays on day of analysis. In case of HPLC analysis, extracts were kept in deep freezer to minimize the change of samples.

L 209-220 Please provide chromatograms of standards and sample extracts. 

Ans) Thank you for your suggestion. We provide HPLC chromatograms in the revised manuscript. Please see Figure 3 in the revised manuscript.

As previous comments, this section must be restructured to show the data better. 

Ans) Thank you for your suggestion. We restructured and rewrite this section as you suggested. Please see line 237-250 in the revised manuscript.

Based on the assays and HPLC results which HHPE parameters were optimized? The data has not been well stated and discussed. 

Ans) Thank you for your suggestion. HPLC results were not used to find the best HHPE conditions because we do not sure which components is important. Consequently, extraction yield and antioxidant activity were considered to find the best HHPE conditions. We provide the HPLC results for which component is extracted with changing HHPE parameters.

L 246-255 present only results in this section and leave discussion in the next section 

Ans) Thank you for your suggestion. It was moved the discussion part. Please see line 330-333 in the revised manuscript.

Table 3 

How many raw data were used for the statistical tests? Explain whether the data were assessed for any normal distribution and outliers?  

Ans) Thank you for your suggestion. Every experiments were repeated three times with three replicates. Consequently, at least 9 raw data were used in each experiment. All data were normally distributed and there were no outliers in this work.

Figure captions more description is needed to understand the data presented in the bar charts. Control samples were not mentioned in the methods and this sample was displayed in figure 3, please explain and confirm. 

Ans) Thank you for your suggestion. We rewrite the figure captions for better understanding. In Figure2, we would like to see the combination effects of puffing and HHPE and non-puffed with no HHPE (0.1 MPa) one is the control sample. We changed Figure 2 and rewrite this in the revised manuscript. Please see line 110-112 in the revised manuscript.

Discussion 

L 269-277 too much background information is not required here. 

Ans) Thank you for your suggestion. We removed some background information. Please see line 305-308 in the revised manuscript.

L 283-284 Justify the claim by comparing data obtained from the study. Please avoid any speculations when the data did not provide the clear-cut conclusion. Any references using the same turmeric as matrix? 

Ans) Thank you for your suggestion. In HPLC analysis of HHPE turmeric the amounts of major curcuminoids (CUR, DMC and BDMC) were greatly affected by ethanol concentration indicating their low water solubility. On the contrary, extraction yield was the highest in the absence of ethanol. These two factors make this conclusion. We addressed this in the revised manuscript. Please see line 313-317 in the revised manuscript.

L 297-313 Explain and justify whether the HPLC data collected in the study are only quantitative and would only suggest any degradation, hence not confirming identification of degradation products. How confident are the authors to discuss the points raised in this paragraph? 

Ans) Thank you for your suggestion. Your comment is correct. We followed the reference and changed this paragraph. Please see line 330-338 in the revised manuscript.

Thank you for your warm and detailed comments. We improved the quality of this manuscript by your detailed comments. We appreciate your concern.

Reviewer 3 Report

This manuscript is interesting and very well written. I have minor suggestions for the authors:

Chemical formulas are not written correctly in the M&M section. Please revise the way to write the subscript numbers.

Line 2. Curcuma longa must be in italic in the title.

Line 32. Family name in italic.

Line 132. A space is missing between pH3.6

Line 141. Remove the symbol prior L. Do the authors mean “mL”?. Same comment at lines 150, 151, 155, and 219.

Table 1. Use superscript letters for statistical significance ?

Line 205-206. Verify the font size of the figure title. + Define the abreviations used in the figure (this comment is applicable for all tables and figures of the manuscript)..

Table 2. Please adjust the font size so that all the results are on one line.

Table 2. Something is wrong with this table. First, the title of the first column if not correct. Please move µg/g in the title of the table. Define abbreviations VA, VN, etc. I think that rows and columns should be inverted. Authors could present it in a landscape orientation.

Table 3. Use superscript letters for statistical significance.

Figure 2. Why turmeric in italic?

Table 4. Define abbreviations (VA, VN, etc.). Superscript letters for stats.

Revise references. For example, references 3 and 9. Curcuma longa must be in italic. Ref 22. Staph. aureus in italic.

Author Response

Response to Reviewer 3

This manuscript is interesting and very well written. I have minor suggestions for the authors:

Chemical formulas are not written correctly in the M&M section. Please revise the way to write the subscript numbers.

Ans) Thank you for your suggestion. We correct them accordingly.

Line 2. Curcuma longa must be in italic in the title.

Ans) Thank you for your suggestion. We correct them accordingly.

Line 32. Family name in italic.

Ans) Thank you for your suggestion. We correct them accordingly.

Line 132. A space is missing between pH3.6

Ans) Thank you for your suggestion. We correct them accordingly.

Line 141. Remove the symbol prior L. Do the authors mean “mL”?. Same comment at lines 150, 151, 155, and 219.

Ans) Thank you for your suggestion. There was an error when it convert to template and we fixed them.

Table 1. Use superscript letters for statistical significance ?

Ans) Thank you for your suggestion. We correct them accordingly.

Line 205-206. Verify the font size of the figure title. + Define the abreviations used in the figure (this comment is applicable for all tables and figures of the manuscript).

Ans) Thank you for your suggestion. We correct them accordingly.

Table 2. Please adjust the font size so that all the results are on one line.

Ans) Thank you for your suggestion. We correct them accordingly.

Table 2. Something is wrong with this table. First, the title of the first column if not correct. Please move µg/g in the title of the table. Define abbreviations VA, VN, etc. I think that rows and columns should be inverted. Authors could present it in a landscape orientation.

Ans) Thank you for your suggestion. We correct them accordingly.

Table 3. Use superscript letters for statistical significance.

Ans) Thank you for your suggestion. We correct them accordingly.

Figure 2. Why turmeric in italic?

Ans) Thank you for your suggestion. We correct them accordingly.

Table 4. Define abbreviations (VA, VN, etc.). Superscript letters for stats.

Ans) Thank you for your suggestion. We correct them accordingly.

Revise references. For example, references 3 and 9. Curcuma longa must be in italic. Ref 22. Staph. aureus in italic.

Ans) Thank you for your suggestion. We correct them accordingly.

Round 2

Reviewer 2 Report

Title. 

Authors are advised to revise the title to fully reflect the results and conclusion obtained from the study. 

The HPLC chromatograms reveal that the peaks have poor resolution and baseline separation, considering this drawback, the qualitative and quantitative data could be inconclusive. Standards and retention times used as reference for obtaining the data are not sufficient given the HPLC method used in the study. 

Introduction 

L 32 delete ‘in many kinds of processed foods’, because turmeric is also used fresh in many countries for fresh food preparations/cooking. 

L 32-41 Information obtained from the literature must be in ‘present tense’ 

L 63 and L 65 insert citations 

L consider to add  ‘ extraction of turmeric ‘ for clarity 

Methods 

L 86 The text ‘Repetitive HHPE was carried out for 3 independent observations’ could be misleading as there were not sufficient data were collected, thus it is recommended to express the sentence differently. 

L 99 specify the pressure setting when the extraction time was evaluated 

L 100 and 101 specify the constant variable for clarity 

L 109-110 rewrite and replace the sentence ‘HHPE was applied at previously found best conditions’  

L 115 insert reference for the gravimetric method used for moisture determination. 

L 138 what is the molarity and pH of the PBS? 

Section 2.5 Which instrument was used for spectrophotometric measurements of the assays?  

Section 2.6 What was the molarity of Na2CO3?  

Which samples were prepared and how many replicates (n=3) must be briefly described in 2.5 and 2.6. 

L 163 provide details of the HPLC column 

L 180 statistically significant 

Section 2.7 provide the reference for the HPLC method. Was the HPLC method developed and optimized in-house rather than adopting from a published method? Please clarify. What are the LOD and LOQ? Was the method validated for quantitative analyses of the compounds of interests? 

Results 

Section 3.1 Please describe from what type of turmeric samples were the data obtained. According to the method section, this experiment used only dried ground turmeric. 

Caption of Table 1. which type of turmeric? 

Section 3.2 The results presented here are also from dried turmeric.  

L 248 Leave the discussion of results in the next section 

Figures: Please place the alphabets (a, b, c and d) from the left to the right in all bar charts 

L 276-282 Please move texts to the method and discussion sections and present only results in this section 

Thank you for showing the chromatograms, though they are cropped and do not show the full run. 

The peaks of the standards were well resolved, although peaks of the samples don’t have good baseline separation. 

Explain and justify whether samples must undergo another treatment (e.g SPE or other methods) to remove any matrix interferences. 

Some peaks have low signal intensities, please provide the LOD and LOQ of the methods to support the data presented in the study. 

What are the LOD and LOQ for VA? 

What was the reason as why the VN and FA peaks are not separated? How were the compounds confidently identified given poor baseline? 

Discussion 

L 305-329 Results are to be discussed in this section, and repetitive presentation of the results should be avoided. 

L 330-338 Discussion is to be limited only to what is relevant with the results and out of the scope and conjecture should be prevented here. 

The final conclusion must be improved by focusing on what was achieved from the study. 

Author Response

2nd Response to reviewer 2

Authors are advised to revise the title to fully reflect the results and conclusion obtained from the study. 

Ans) The authors appreciate for the suggestion, and the title is revised as “Enhanced antioxidant capacity of puffed turmeric (Curcuma long L.) by high hydrostatic pressure extraction (HHPE) of active compounds”.

The HPLC chromatograms reveal that the peaks have poor resolution and baseline separation, considering this drawback, the qualitative and quantitative data could be inconclusive. Standards and retention times used as reference for obtaining the data are not sufficient given the HPLC method used in the study. 

Ans) Thank you for your suggestion. We added a reference, which was recently developed and published work, in the revised manuscript. We rewrite the HPLC methodology. Please see line 162-173 in the revised manuscript. All the seven standard compounds were obtained from Sigma, which were available for HPLC analyses.

Introduction 

L 32 delete ‘in many kinds of processed foods’, because turmeric is also used fresh in many countries for fresh food preparations/cooking. 

Ans) Thank you for your suggestion. We removed it as you suggested. Please see line 33 in the revised manuscript.

L 32-41 Information obtained from the literature must be in ‘present tense’ 

Ans) Thank you for your suggestion. We corrected it as you suggested. Please see line 33-41 in the revised manuscript.

L 63 and L 65 insert citations 

Ans) Thank you for your suggestion. We added references as you suggested. Please see line 64 and 66 in the revised manuscript.

L consider to add  ‘ extraction of turmeric ‘ for clarity 

Ans) Thank you for your suggestion. We added references as you suggested. Please see line 67 in the revised manuscript.

Methods 

L 86 The text ‘Repetitive HHPE was carried out for 3 independent observations’ could be misleading as there were not sufficient data were collected, thus it is recommended to express the sentence differently. 

Ans) Thank you for your suggestion. We corrected it as you suggested. Please see line 87-89 in the revised manuscript.

L 99 specify the pressure setting when the extraction time was evaluated 

Ans) Thank you for your suggestion. We corrected it as you suggested. Please see line 100 in the revised manuscript.

L 100 and 101 specify the constant variable for clarity 

Ans) Thank you for your suggestion. We corrected it as you suggested. Please see line 101 and 102 in the revised manuscript.

L 109-110 rewrite and replace the sentence ‘HHPE was applied at previously found best conditions’  

Ans) Thank you for your suggestion. We replaced it as you suggested. Please see line 110-111 in the revised manuscript.

L 115 insert reference for the gravimetric method used for moisture determination. 

Ans) Thank you for your suggestion. We added a reference as you suggested. Please see line 117 in the revised manuscript.

L 138 what is the molarity and pH of the PBS? 

Ans) Thank you for your suggestion. The molarity and pH of PBS were 0.15 M NaCl and 0.01 M KH2PO4 as well as pH 7.4.

Section 2.5 Which instrument was used for spectrophotometric measurements of the assays?  

Ans) Thank you for your suggestion. We used a 96-well plate using a microplate reader (Bio-Rad, Hercules, CA, USA). We added this instrument in the revised manuscript. Please see line 133-134 in the revised manuscript.

Section 2.6 What was the molarity of Na2CO3?  

Ans) Thank you for your suggestion. The molarity of Na2CO3 is 0.66 M.

Which samples were prepared and how many replicates (n=3) must be briefly described in 2.5 and 2.6. 

Ans) Thank you for your suggestion. In case of section 2.5 and 2.6, we determined all samples listed in this manuscript. All measurements were repeated three times with three replicates as mentioned in section 2.8. We added this in the revised manuscript. Please see line 128-129, 155-156 and 176 in the revised manuscript.

L 163 provide details of the HPLC column 

Ans) Thank you for your suggestion. We provided the details of the HPLC column in the revised manuscript. Please see line 167-168 in the revised manuscript.

L 180 statistically significant 

Ans) Thank you for your suggestion. We correct it as you suggested. Please see line 183 in the revised manuscript.

Section 2.7 provide the reference for the HPLC method. Was the HPLC method developed and optimized in-house rather than adopting from a published method? Please clarify. What are the LOD and LOQ? Was the method validated for quantitative analyses of the compounds of interests? 

Ans) Thank you for your suggestion. We added a reference, which was recently developed and published work, in the revised manuscript. We rewrite the HPLC methodology. Please see line 163-173 in the revised manuscript. All the seven standard compounds were obtained from Sigma, which were available for HPLC analyses.

Results 

Section 3.1 Please describe from what type of turmeric samples were the data obtained. According to the method section, this experiment used only dried ground turmeric. 

Ans) Thank you for your suggestion. We determined the extraction yield of ground turmeric in this section and correct it as you suggested. Please see line 187 in the revised manuscript.

Caption of Table 1. which type of turmeric? 

Ans) Thank you for your suggestion. We determined the extraction yield of ground turmeric in this section and correct it as you suggested. Please see line 197 in the revised manuscript.

Section 3.2 The results presented here are also from dried turmeric.  

Ans) Thank you for your suggestion. You are right. We determined the antioxidant activity and TPC of ground turmeric in this section and correct it as you suggested. Please see line 201 in the revised manuscript.

L 248 Leave the discussion of results in the next section 

Ans) Thank you for your suggestion. We move this part to discussion section. Please see line 317-319 in the revised manuscript.

Figures: Please place the alphabets (a, b, c and d) from the left to the right in all bar charts 

Ans) Thank you for your suggestion. We corrected it as you suggested.

L 276-282 Please move texts to the method and discussion sections and present only results in this section 

Ans) Thank you for your suggestion. We corrected it as you suggested. Please see line 337 in the revised manuscript.

Thank you for showing the chromatograms, though they are cropped and do not show the full run. 

The peaks of the standards were well resolved, although peaks of the samples don’t have good baseline separation. 

Explain and justify whether samples must undergo another treatment (e.g SPE or other methods) to remove any matrix interferences. 

Some peaks have low signal intensities, please provide the LOD and LOQ of the methods to support the data presented in the study. 

What are the LOD and LOQ for VA? 

Ans) Thank you for your suggestion. According to the chromatograms in Figure 3, the baseline does not seem to be good. However, all the baselines of the chromatograms were not actually unstable when enlarged. Curcumin peak is relatively, much larger than those of the four degradation products including vanillic acid, vanillin, ferulic acid, and 4-vinyl guaiacol. Therefore, the peaks of the degradation products in the chromatogram were smaller than the peak of curcumin. Whenever the samples were analyzed using HPLC in this study, the mixture of the seven standard compounds were injected as an external standard; the matched peaks in the samples were clearly identified by comparison with the peaks of the seven standards. Please check the manuscript.

What was the reason as why the VN and FA peaks are not separated? How were the compounds confidently identified given poor baseline? 

Ans) Thank you for your suggestion. As answered above, VN and FA of all the samples were separated, enabling identification and quantification of VN and FA. Whenever the samples were analyzed using HPLC in this study, the mixture of the seven standard compounds were injected as an external standard; the matched peaks in the samples were clearly identified by comparison with the peaks of the seven standards. Please check the manuscript.

Discussion 

L 305-329 Results are to be discussed in this section, and repetitive presentation of the results should be avoided. 

Ans) Thank you for your suggestion. We corrected it as you suggested. Please see line 313-328 in the revised manuscript.

L 330-338 Discussion is to be limited only to what is relevant with the results and out of the scope and conjecture should be prevented here. 

Ans) Thank you for your suggestion. We corrected it as you suggested. Please see line 348-353 in the revised manuscript.

The final conclusion must be improved by focusing on what was achieved from the study. 

Ans) Thank you for your suggestion. We corrected it as you suggested. Please see line 357-362 in the revised manuscript.
